# Error Correcting by Agreement Checking for Adversarial Robustness against Black-box Attacks

## Abstract

Drawing inspiration from the vulnerability of the initial feed-forward phase of biological perception in humans and primates to adversarial attacks, we propose a novel defense strategy named Error Correcting by Agreement Checking (ECAC). This strategy is designed to mitigate realistic *black-box* threats where attackers don't have full access to the model. We exploit the fact that natural and adversarially trained models rely on distinct feature sets for classification. Notably, naturally trained models retain commendable accuracy against adversarial examples generated using adversarially trained models. Leveraging this disparity, ECAC moves the input toward the prediction of the naturally trained model unless it leads to disagreement in prediction between the two models, before making the prediction. This simple error correction mechanism is highly effective against leading SQA (Score-based Query Attacks) black-box attacks as well as decision-based and transfer-based black-box attacks. We also verify that, unlike other black-box defense, ECAC maintains significant robustness even when adversary has full access to the model. We demonstrate its effectiveness through comprehensive experiments across various datasets (CIFAR and ImageNet) and architectures (ResNet as well as ViT).

## 1 Introduction

Since the advent of adversarial attacks (Szegedy et al., 2014), the field has seen an arms race between adversarial defenses and attacks. Defenses based on adversarial training, which incorporates adversarially crafted inputs during training, such as SAT (Standard-Adversarial Training) (Madry et al., 2018), TRADES (Zhang et al., 2019), and AWP (Wu et al., 2020), have withstood the test of time. However, robust accuracy still needs improvement for reliable deployment.

In realistic scenarios, attackers lack complete access to models, making black-box defense a practical choice. Given access to the model confidence scores, attackers can deploy query-efficient SQA attacks. When only the final prediction classes are accessible, decision-based attacks become a feasible strategy. Since these attacks require a large number of queries to the model, many SQA defense methods ignore them, considering it as impractical. However these attacks can be amplified when combined with techniques like transfer attacks.

In situations where attackers have access to similar data, they are capable of training surrogate models to execute powerful transfer attacks. Our work focuses on these practical challenges, proposing a defense that works well against all these black-box attacks.

We believe it is challenging for simple feed-forward networks to achieve adversarial robustness because they lack an error correction mechanism (Hawkins & Sandra, 2004) as utilized by biological perception.

Table 1: Transfer accuracy of adversaries generated by different models. (ResNet-18, CIAFR-10, PGD-100 attack). Columns show models used in crafting the adversaries.

|        | Nat'  | SAT   | TRADES | MART  |
|--------|-------|-------|--------|-------|
| Nat'   | 00.00 | **71.27** | **72.50** | **75.97** |
| SAT    | **82.74** | 51.61 | 62.22  | 64.19 |
| TRADES | **82.67** | 62.53 | 52.94  | 66.23 |
| MART   | **78.42** | 59.01 | 61.03  | 54.87 |

Elsayed et al. (2018) showed that when humans were forced to make classification under time-limited settings ($\approx 74$ ms), i.e., when they likely cannot use the error correction mechanism in their perception, adversarially crafted images also fool them. Based on this observation, we propose devising an error correction mechanism to boost the adversarial robustness of a trained model.

One main challenge is to identify an appropriate error signal. We propose to use the disagreement between the predictions by adversarially trained and naturally trained models as a substitute error oracle. Zhang & Zhu (2019) showed that adversarially-trained models are shape-biased, while naturally trained models are texture-biased, i.e., the two models mostly use different features. We observed that these different features, namely shape and texture, are orthogonal to each other, making it particularly challenging to transfer adversarial examples between them. We illustrate this in Table 1 for three adversarially trained models. For adversarial examples crafted using adversarially trained models, the naturally trained model gives much higher accuracy than any other model on the same network, and vice versa.

Inspired by these observations, we devise a simple error correction mechanism to boost robustness in black-box settings. We first record the prediction made by the naturally trained model (referred to as $weakM$). Then we perturb (or $nudge$) the input towards that prediction by a 1-step targeted PGD attack, employing both the naturally and adversarially trained models in parallel, as described later (referred to as $jointM$). If the prediction made by the adversarially trained model (referred to as $strongM$) on the nudged input matches our initial prediction, we output that prediction, otherwise, we output the prediction by $strongM$ on the original input. We always output the logits of $strongM$. Figure 1 depicts the overall architecture. The algorithm is presented in Section 3.2. Our method operates under the assumption that if the input is perturbed towards the correct class, the two models will more readily agree on their predictions.

To the best of our knowledge, we are the first to exploit the naturally trained model

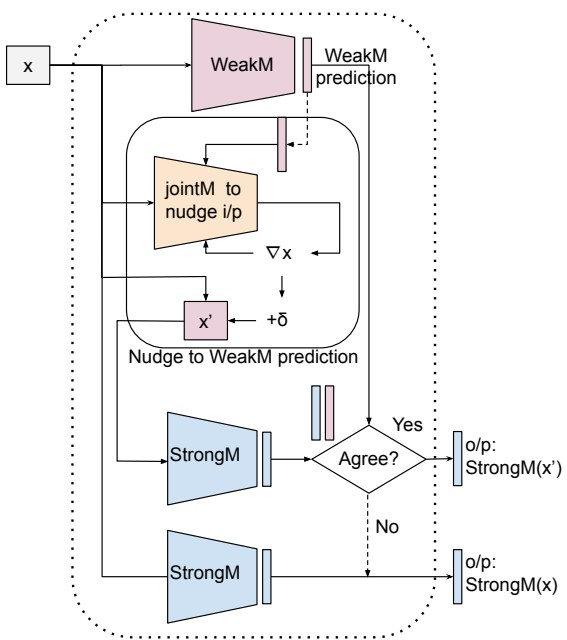

Figure 1: ECAC Architecture. *WeakM* refers to a naturally trained model, *StrongM* refers to an adversarially trained model, and *jointM* refers to when the models are in parallel.

to amplify the robustness of the adversarially trained model. Our contributions are summarized as follows:

- Inspired by biological perception, we argue for the necessity of error correction for adversarial defense. We introduce a novel Error Correcting by Agreement Checking (ECAC) method for defense against black-box attacks.
- We experimentally show that ECAC significantly enhances the robustness of adversarially-trained models against realistic attacks: SQA, decision-based, transfer, and also adaptive black-box attacks. Further, We also verify that, unlike other black-box defense, ECAC maintains significant robustness even when adversary has full access to the model.

## 2 BACKGROUND AND RELATED WORK

### 2.1 PRELIMINARIES

We work with $K$ class classifier $f$, parameterized by $\boldsymbol{\theta}$, which maps the input $x_i$ from the input space $\mathcal{X}$ to its corresponding class $y_i$. For each input, the model outputs a score for each class $c$ as

$f_c(x_i, \boldsymbol{\theta})$, called logits. The class with the highest logit is considered the predicted class. We denote it as $y_{pred_i} = \arg\max_c f_c(x_i, \boldsymbol{\theta})$. In this text, we refer to naturally trained model as $weakM$ as they do not exhibit robustness against adversarial attacks. An adversarially trained model is referred to as $strongM$. We refer to the predictions made by the two models $strongM$ and $weakM$ as $y_{strongM_i}$ and $y_{weakM_i}$ respectively. When we perturb (or $nudge$) the input as part of our defense, we use $weakM$ and $strongM$ in parallel by summing the loss (cross-entropy) from the two models and refer to it as $jointM$. In the adversarial setting, an adversary perturbs the input $x_i$ to $x_i'$ to cause misclassification. The perturbation is limited to the neighborhood of the original input defined by the ball $B_\epsilon[x_i] = \{x_i' : \|x_i' - x_i\|_p < \epsilon\}$ where $\|.\|_p$ denotes the $\ell_p$ norm, and $\epsilon$ is the perturbation budget set to $8/255$ for CIFAR-10 and $4/255$ for ImageNet. Since the attacks in the $\ell_\infty$ norm are much stronger, we limit ourselves to it.

## 2.2 ADVERSARIAL ATTACKS AND DEFENSES

Due to space constraints, we present the prominent adversarial attacks and defense methods in Appendix D. We want to point out that most black-box defenses do not consider decision-based and transfer attacks. We believe this gives a false sense of security, as an adversary can easily bypass such defenses by training a surrogate model. Szegedy et al. (2014); Sitawarin et al. (2024) further underscore the relevance of defenses against transfer attacks.

**Related work:** ECAC (ours) is an adaptive defense. Defenses that alter the input or model parameters or both during defense can be categorized as adaptive defense. ECAC alters the input by nudging it toward the expected correct class. Several existing works also nudge the input as our method, although for different reasons. Wu et al. (2021) nudges the input such that the cross-entropy summed over all classes is maximized. However, as noted by Croce et al. (2022), this may often reduce the accuracy when the input is near the decision boundary. Shi et al. (2021) nudges the input using self-supervised signals while we leverage the disagreement between the $strongM$ and $weakM$, i.e., we use the signal from $weakM$ as well. Tao et al. (2022) and Li et al. (2023) use input nudging during the training time (and are not adaptive defenses).

Qin et al. (2021) showed that simply adding Gaussian noise, of sufficient magnitude, provides nontrivial accuracy for several SQA attacks. This defense is popularly known as RND. Chen et al. (2022) have devised an adaptive method, dubbed AAA, most similar in spirit to ours, for defense against black-box SQA. Instead of outputting true logits, they post-process it such that SQA locally observes an increase in logits when it really decreases. Thus they are able to successfully mislead the black-box SQA attackers. However, their method is limited to defense against SQA and does not increase the correct decision area around an input, thus it remains exposed to the transfer attacks.

## 3 METHODOLOGY

### 3.1 MOTIVATION FROM BIOLOGICAL PERCEPTION: NEED FOR ERROR CORRECTION

Despite significant research in making neural network models robust, their robustness remains decisively inferior to biological perception. While adversarially trained models seem to extract human-interpretable features (Zhang & Zhu, 2019), it has been shown that even adversarially trained models are brittle to real-world transformations like rotation and translation (Engstrom et al., 2019). Thus, we ponder:

> *Can feed-forward neural networks be adversarially robust, or are they intrinsically vulnerable?*

Elsayed et al. (2018) showed that if adversarially crafted images are shown for a very brief moment (about 71ms), they can even fool humans. However, humans easily predict the true class with ample time ($\sim 2$ seconds). They attributed it to the top-down and lateral connections in human brains.

Guo et al. (2022) showed that artificial neural networks might already be more robust in representing the input than biological ones. They compared the IT layer in monkeys with the feature layer of ResNet-50 and observed that adversarially trained ResNet-50's encoding is less sensitive to adversarial perturbation. Further, the attack on the biological system was not done using a white-box but a weaker black-box transfer attack, indicating that they are even more sensitive to the adversar-

ial attack. However, biological brains do not seem to be fooled by adversarial attacks. The paper attributes this to some error-correcting mechanism of the brain not fully understood.

Inspired by the above observations, we propose a simple way to correct the input using backward propagation, described in detail in the following section. On the surface, it may crudely look similar to the suggested top-down mechanism of the neocortex (Hawkins & Sandra, 2004), but biological perception is much more sophisticated, and the comparison will not be apt.

### 3.1.1 OUR INSIGHT

While making a prediction, if an oracle could point us toward the correct class, we could nudge the input toward that class, thereby undoing the adversarial perturbations. We realize we could leverage the disagreement between the *strongM* (adversarially trained model) and *weakM* (naturally trained model) predictions as an oracle.

As shown in Table 1, transferring an attack from an adversarially trained model to a naturally trained one is difficult. This is because the two models use different features. Naturally trained models often rely on smaller magnitude features (like texture) that are abundantly present in an image. This was demonstrated by Zhang & Zhu (2019), where they showed that if an image is divided into tiles, and the tiles are shuffled, naturally trained models still manage to predict the true class, while adversarially trained models fail. However, texture-like features are easier to remove by adversarial attacks. Without adversarial training, the model has no incentive to learn better features (Tsipras et al., 2018). On the contrary, adversarially trained models learn to use shape-aware features and thus become indifferent toward the texture. We found that these two models can be combined to get an overall more robust model.

We work with the assumption that when the two models agree, their agreed prediction tends to be correct. This is based on the intuition that the models are trained to make the correct predictions and the chance of both *strongM* and *weakM* predicting the same wrong class is low. We experimentally validated this assumption for CIFAR-10 using the ResNet-18 model. We found that when an adversary example is both crafted and nudged using *jointM*, then nudging towards the correct class increases the agreement between the two models' predictions from $34.98\%$ to $49.20\%$, while if they are nudged towards a random class, the agreement decreases to $16.99\%$. Further analysis of how and when this assumption holds is given in Appendix A in the supplement. Based on this assumption, we design our defense as explained next.

### 3.2 ERROR CORRECTING BY AGREEMENT CHECKING

We present the ECAC algo in Algorithm 1. To nudge the input, we use a targeted PGD attack as shown in equation 1. Here the loss $l(\cdot)$ is the summation of the (cross-entropy) loss incurred by *weakM* and *strongM*. The algorithm has the following steps:

- We first note the logit returned by *weakM* on the input $x_i$. We term it as $logitW$ (Step 1), with prediction: $y_i^o$.

$$x_i'^{t+1} \leftarrow \prod_{B_\epsilon[x_i]} (x_i'^t - s\_size \cdot sign(\nabla_{x_i'^t} l(x_i'^t, y_i^o))), \quad (1)$$

- Then using the above equation (1), we nudge the input towards $logitW$ prediction $y_i^o$ to get $x_i'$. (Step 2).

- We then calculate the logit returned by *strongM* on the nudged input $x_i'$ and term it $logitS'$. (Step 3).

- If prediction made by $logitS'$ matches $logitW$, we return $logitS'$ as final output. (Step 4 to 6).

- However, on disagreement, we discard the nudging and simply return the logit as calculated by *strongM* on the original input $x_i$. (Step 7).

---

**Algorithm 1** Error Correcting by Agreement Checking

**Inputs:** $x_i$, *strongM*, *weakM*, $s\_size$
**Output:** Prediction-logits for the input $x_i$.

1: $logitW \leftarrow weakM(x_i)$.
2: Get $x_i'$ by nudging $x_i$ toward $argmax(logitW)$ using Equation (1) by $s\_size$ for 1 step.
3: $logitS' \leftarrow strongM(x_i')$.
4: **if** $argmax(logitS') == argmax(logitW)$ **then**
5:    $return\ logitS'$.
6: **end if**
7: $Return\ strongM(x_i)$.

---

## 3.3 ADAPTIVE DEFENSE BY ECAC, ON SQA ATTACKS

ECAC dynamically shifts the decision boundary, making it harder for attacks to reach it. Further, if an attacker manages to fool one model, say *strongM* (as we always output its logit), and if *weakM* does not get fooled, *weakM* would nudge the input towards the correct prediction, thwarting the attack. This is illustrated in Figure 2.

Consider the case where a natural input $x_i$ is correctly classified, point (a) in the figure. An adversarial attack would perturb the input in an attempt to misclassify it. If the two models agree on their prediction, i.e., before point (b) in the figure, the output would be $strongM(x_i')$. In this case, also, the logit would be returned only after nudging the input towards the agreed class (not shown in the figure to avoid clutter). However, when the models start disagreeing, we have two cases: either *weakM* will make the correct prediction (case A) or not (case B).

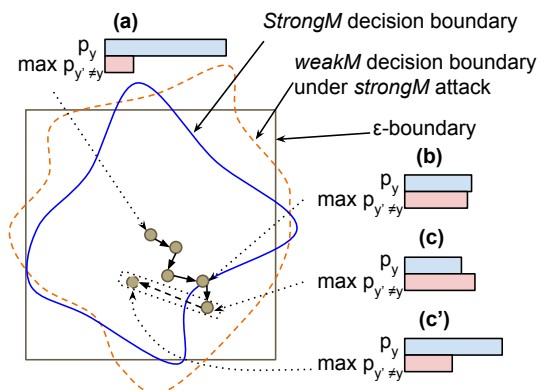

We show case A, point (c) in the figure. Here ECAC would nudge $x_i'$ towards *weakM* prediction by $\delta_i$ (point (c') in the figure). Since nudging is done using *jointM* toward the prediction made by *weakM*, we have $\delta_i \propto sign(\nabla_{x_i'}(l_{ce}(strongM(x_i'), y_{weakM_i}) + l_{ce}(weakM(x_i'), y_{weakM_i})))$. If the two models agree at point (c'), the output is $strongM(x_i' + \delta_i)$. Notice that although *strongM* makes an incorrect prediction for point (c), ECAC can fix it and output logits for a point well within the decision boundary, aka point (c'), of *strongM*. This effectively increases the decision boundary for *strongM*. If, instead, the two models disagree at $x_i'+\delta_i$ (disagreement case not shown in the figure), we would discard the nudging and output prediction as made by *strongM*.

Figure 2: Illustration of adaptive defense by ECAC. See the text for details.

For case B, when *weakM* makes an incorrect prediction, we rely on robustness of *strongM*. Since it is hard to fool *strongM*, the nudging done in the incorrect direction would, often, not be enough for *strongM* to cause misclassification, especially under realistic black-box scenario. We discuss fraction of each cases occurring for natural samples for CIFAR-10 in Appendix B.1.

### 3.3.1 ROBUSTNESS TO BLACK-BOX ATTACKS

Owing to its adaptivity, ECAC is especially robust to black-box attacks. SQA attacks estimate the vulnerability of a sample based on the score assigned to the true class by the model. These attacks iteratively perturb an input region, progressively decreasing the models' confidence in the true class. A perturbation for an iteration is discarded if it fails to lower the true class probability.

We demonstrate this error correction for the Square attack in Figure 3. For the demonstration we used smaller, ResNet-18, models trained on CIFAR-10. *strongM* is trained with $\epsilon = 8/255$. We selected

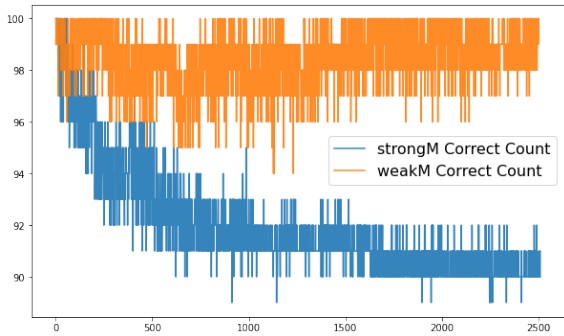

Figure 3: Effectiveness of ECAC on the Square attack. The Square attack is able to successfully attack these 100 samples, individually, for both naturally trained (*weakM*) and adversarially trained (*strongM*) models. However, Square attack fails when the same models are used in ECAC. We can see that *weakM* is able to fix the error in prediction made by *strongM*, and vice versa.

100 samples, such that the Square attack is able to fool the *strongM* in 2500 iterations, while ECAC makes the correct prediction. For each iteration, we measure the number of samples (out of these 100) which are correctly predicted by both *strongM* and *weakM*. As shown in the Figure 3, we notice that *strongM* makes significantly more errors as compared to *weakM*, and in each iteration *weakM* fixes the prediction made by *strongM* and vice-versa, demonstrating the efficacy of ECAC. We can clearly notice that after around 500 to 1000 iterations, the Square attack seems to be stuck in a loop. It seems that the perturbed input is near the decision boundary for *strongM* and the Square attack is often able to perturb it past that as well. However, the error correction mechanism of ECAC, with the help of *weakM*, keeps fixing it. Thus the Square attack keeps discarding the perturbations made by it. We show the accuracy on Square and several other black-box attacks in Table 2.

Hard-label attacks, like the popular SPSA attack, also get fooled similarly. SPSA first estimates the gradients using finite difference estimates in random directions, then uses that gradient to perturb the input. However, a random perturbation will rarely cause both models to fail, simultaneously, in their prediction, and ECAC would likely predict the correct class. Thus the SPSA attack would fail in estimating a perturbation direction. This is evident from the results in Table 4, where ECAC exhibits significant robustness against SPSA accuracy. We also demonstrate the efficacy of ECAC on the popular RayS (Chen & Gu, 2020) attack.

We further point out that for transfer attacks, unlike other black-box defenses, ECAC maintains significant accuracy. This is primarily because we use an adversarially trained model as part of the defense. As pointed out by Tsipras et al. (2018), naturally trained models do not have an incentive to learn robust features, and it is easier to perturb non-robust features to make such defenses susceptible to transfer attacks. We discuss transfer as well as adaptive attacks in Section 4.2. Finally, even in a scenario where an adversary has full knowledge and access to the model (i.e., a white-box attack), ECAC would still demonstrate significant robustness. This is because we consistently output strongM logits, ensuring that the accuracy remains lower-bounded by $strongM(x + \epsilon + \delta)$, where $\epsilon$ is the allowed perturbation budget and $\delta$ represents the adjustments made by ECAC.

## 3.4 DESIGN CHOICES

Given the observation that the two models use different sets of features, and only *strongM* is adversarially robust, an intuitive design choice could be to nudge the input only when the models start disagreeing on their prediction. This way, only *strongM* is exposed to the attacker till ECAC makes the correct prediction. This approach boosts robustness but is nearly $10\%$ less effective against several SQA attacks, unless we fix the perturbation continuously. Therefore, we chose to continuously correct the input and discard the nudging if it doesn't lead to agreement.

We always use *strongM* to output the logits. This way *weakM* confidence is kept hidden from the attacker. Further, since it is much easier to fool the *weakM* model, if the two models disagree on their prediction, we rely on the prediction made by *strongM*. We observe that if we use *weakM* instead of *strongM* to output the logits, than the Square attack accuracy, for 100 iterations, on the first $1K$ testset samples of CIFAR-10 drops from $83.50\%$ to $71.80\%$.

We also needed to select an appropriate model for nudging the inputs. To ensure the input remains within the correctly classified region for both the models (when it is being attacked), we nudge using both in parallel. Specifically, we use *jointM*, which combines *weakM* and *strongM*, to perform the nudging.

To nudge the input, values for two parameters are needed: a) $n\_steps$: the number of PGD steps to nudge the input toward a class, and b) $s\_size$: the size of each such step. $n\_steps \times s\_size$ determines the amount of nudging. Since *strongM* is trained with a perturbation budget of $\epsilon = \frac{8}{255}$ and for *weakM*, this value is 0, the right amount of nudging should be somewhere in between, i.e., around $\epsilon/2$. We observed that the effect of doing multiple small steps could be approximated with a single large step. Thus we set the value of $n\_steps$ to 1 (and remove it as a parameter). Further, as shown in the table 7 in section 4.3.1, we observed that the value of $s\_size$ leads to a tradeoff between transfer and black-box attacks. By experimenting on CIFAR-10 for ResNet-18, we freeze on a value of 0.02 for $s\_size$. Thus we have $s\_size = \delta$. For a slightly higher value, ECAC exhibits slightly lower accuracy for transfer attacks but slightly higher accuracies for black-box attacks and vice versa.

Table 2: ECAC performance compared to baselines on SQA attacks for the CIFAR-10 dataset with a perturbation budget of $\ell_\infty = \frac{8}{255}$ (queries = 100/2500). WideResNet-28-10 is used for all models.

| Defense Methodology | Nat' Acc' | Accuracy on SQA Attack (# queries = 100/2500) | | | | |
|---|---|---|---|---|---|---|
| | | Square | SignHunter | SimBA | NES | Bandit |
| Undefended | 94.78 | 39.7/00.2 | 42.3/00.0 | 73.5/35.6 | 68.8/05.0 | 49.9/01.3 |
| SAT | 85.83 | 76.9/60.5 | 74.9/56.6 | 84.1/80.4 | 83.3/75.4 | 78.7/66.2 |
| RND | 91.05 | 60.8/49.1 | 61.0/47.8 | 76.4/64.3 | 86.2/68.2 | 70.4/41.6 |
| AAA-Linear | **94.84** | 83.4/79.8 | 84.2/83.0 | 86.4/84.5 | 84.6/71.0 | 86.7/82.8 |
| ECAC-SAT | 90.30 | 85.7/84.3 | 80.5/79.4 | 86.0/83.6 | 87.3/73.2 | 85.0/81.9 |
| ECAC-TRADES | 91.65 | 87.4/85.8 | 81.0/79.0 | 86.6/85.8 | 87.9/74.7 | 85.7/83.0 |
| ECAC-AWP | 90.00 | 86.9/85.0 | 79.7/77.7 | 86.4/84.9 | 87.0/75.1 | 85.1/82.5 |
| ECAC-AWP_E | 91.80 | 87.8/87.5 | 83.8/82.5 | 88.0/85.8 | 88.4/77.5 | 86.0/83.7 |
| ECAC-WANG23 | 94.40 | **91.4/90.9** | **87.2/85.6** | **91.0/89.5** | **92.1/81.6** | **89.8/88.0** |

For ImageNet, since we use a perturbation budge of $4/255$, i.e., half of what has been used for CIFAR-10, we set $s\_size = 0.01$ for ImageNet. We obtained good performance for all datasets and models using this parameter.

# 4 EXPERIMENTS

## 4.1 SETUP

We evaluated ECAC on CIFAR-10 and ImageNet datasets. For fine-tuning and ablation, we used ResNet-18. In line with our baseline AAA-linear (Chen et al., 2022), we reported the result using WideResNet-28-10 for CIFAR-10 and WideResnet-50 for ImageNet. In Appendix E, we provide the source of these models along with the training details for the models trained locally.

For Square attacks, we used the official implementation of AA (https://github.com/fra31/auto-attack). In line with our baseline, for the Square attack we set $p\_init = 0.05$ (the fraction of pixels changed on every iteration). Auto Attack uses a value of 0.8 for this parameter. We note that changing this parameter considerably lowers the AAA-Linear accuracy. For the rest of the SQA attacks, we used the same parameter as used by AAA-Linear and provide the details in the supplement. We did 100 iterations for SPSA with a perturbation size of 0.001, a learning rate of 0.01, and a total of 128 samples for each gradient estimation. For RayS, we used the official implementation (https://github.com/uclaml/RayS) with 10K (and 1K) queries.

## 4.2 RESULTS

In this section, we present the results for several attacks. Due to space constraints, some results have been moved to the Appendix.

**SQA attacks:** Table 2 presents the results for black-box SQA attacks on WideResNet-28-10 models. The parameters used and other details for the attack are provided in the Appendix F. We used the first 1K samples from CIFAR-10. The results for the RND (Qin et al., 2021) attack were borrowed from the AAA paper (Chen et al., 2022).

We first compare ECAC-SAT (also referred simply as ECAC) performance with baselines. For ECAC-SAT, we use basic SAT model (Madry et al., 2018) as $strongM$. For the Square attack, it outperforms all the baselines including AAA-Linear. We discuss the vulnerability of AAA-Linear to the Square attack in the subsequent paragraph. For SignHunter, AAA-Linear performs better than ECAC-SAT. For SimBA, the results are almost similar, despite AAA-Linear's natural accuracy being much higher. ECAC-SAT outperform AAA-Linear on the NES attack; however, for larger queries, its accuracy drops slightly below that of the adversarially trained model. For Bandit, AAA-Linear also performs a bit better.

There are multiple methods to achieve $strongM$, making its selection an additional hyper-parameter. We experimented with TRADES (Zhang et al., 2019), AWP, AWP_E (Wu et al., 2020), and WANG23 (Wang et al., 2023) as $strongM$. Notably, when employing a stronger $strongM$ with the same WideResNet-28-10 architecture, we consistently surpass the state-of-the-art accuracy across all attacks. More information and comparisons of ECAC with the corresponding $strongM$ are provided in Appendix B.2.3.

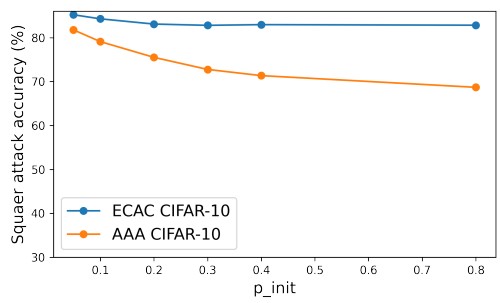

Figure 4: Square attack accuracy of AAA-Linear for different values of $p\_init$ (fraction of pixels changed every iter') for CIFAR-10

For the Square attack, AAA-Linear has used a value of $p\_init = 0.05$ for CIFAR-10 and 0.3 for ImageNet. This is consistent with the original implementation of the Square attack as it was used on a naturally trained model. However, later implementations used a value of 0.8, which is more suitable to attack adversarially robust models. An attacker would likely choose the value that degrades the performance of the model the most. We tested the accuracy of AAA-Linear, using their code, for different values of this parameter and compared it with ECAC accuracy in Fig 4. We note a considerable drop in accuracy for AAA-Linear. For ECAC, however, the accuracy remains almost the same.

We present the results for the ImageNet dataset in Appendix in Table 12. For AAA-Linear, we used their code to run the Square attack. They randomly selected 1k samples, one from each class, which are correctly classified by the naturally trained model. The reported accuracy (for AAA-square) is calculated as the natural accuracy, i.e., $78.48\% \times$ accuracy obtained on the selected 1k samples. For the rest of the results, we selected the first 2k samples from the validation set of ImageNet. We observe that ECAC is able to outperform AAA-Linear for Square and SimBA attacks.

**Transfer attacks:** Most black-box defenses do not account for the possibility of transfer attacks. In the current age of big data and cheap compute, we believe that a resourceful adversary can easily train a surrogate model and transfer the attack to the deployed system (Szegedy et al., 2014).

Table 3: Transfer attack accuracies on CIAFR-10, using ResNet-18 architecture as surrogate Model. We note that, AAA-Linear accuracy drops considerably.

| SurrogM (ResNet-18) | Natural | SAT | AAA | ECAC |
|---|---|---|---|---|
| | (WideResNet-28-10) | | | |
| AT | 73.17 | 64.85 | 73.17 | 69.33 |
| Natural | 16.91 | 85.03 | **16.91** | 79.47 |
| $jointM$ | 15.36 | 78.92 | **15.36** | 72.95 |

It is likely that an adversary will not know the architecture of the deployed model. Therefore, for CIFAR-10, we trained a ResNet-18 model (details of training provided in Appendix E) and used it as our surrogate model. We transferred the attacks crafted using PGD-20 on ResNet-18, a much smaller network, to our target models, all of which use WideResNet-28-10. The results are provided in Table 3. When the adversarially trained model is used to craft the adversary, the AAA defense exhibits the highest robustness, as does the undefended model. This aligns with our expectation, as it is hard to transfer attacks crafted using the adversarially trained model to the naturally trained model. However, if the adversary uses the naturally trained model, which is much easier to train, the accuracy of the AAA defense becomes very poor. Overall, we notice that *ECAC maintains the highest worst-case transfer accuracy (i.e., 69.33%).*

Table 4: ECAC performance compared to baselines, using WideResNet-28-10, on decision-based attacks for CIFAR-10, with the $\ell_\infty$ perturbation of: $\frac{8}{255}$.

| Models | RayS (1K/10K queries) | SPSA |
|---|---|---|
| Undefended | 22.30/00.10 | 00.00 |
| AT | 71.40/59.90 | 62.40 |
| AAA-Linear | 58.50/55.10 | 70.10 |
| ECAC (ours) | **72.00/66.60** | **79.00** |

**Decision-based attacks:** We also tested our method on decision-based attacks (SPSA and RayS). The results are presented in Table 4. Both, our defense and AAA, show good robustness for SPSA, which works by estimating the gradient. However, RayS searches for the decision boundary by starting with a misclassified point (with higher perturbation), and thus is more effective against AAA defense. Since our defense dynamically changes the decision boundary, it is significantly more robust than AAA in both cases. We present the results for ImageNet in Appendix C.2.

**Adaptive attacks (and robustness against white-box attack):** We have demonstrated the performance of ECAC assuming attackers do not know the defense strategy. It is interesting to see how well ECAC holds if attackers know the defense strategy, that is, the ECAC architecture in Figure 1, but do not have access to the individual $strongM$ or $weakM$ models. We assume, like a realistic scenario, an attackers can query the deployed ECAC model. To construct an effective adaptive attack, we utilized surrogate models trained on analogous datasets. For CIFAR-10, we trained ResNet-18 using standard and Madry's [2018] method as our surrogate models.

The essence of adaptive attacks is to exploit a defense's vulnerabilities. For ECAC, potential weaknesses arise when (a) both models agree on an incorrect prediction post-nudging or (b) their predictions diverge and $strongM$ errs on the input. We devised an adaptive attack to exploit both cases. First, we identify $strongM$'s vulnerable class via an untargeted PGD attack with

Table 5: ECAC Accuracy on Adaptive Attack.

| Surrogate Model Accuracy | | | **ECAC** |
| $strongM$ | $weakM$ | ECAC | **Accuracy** |
|---|---|---|---|
| 50.59 | 00.00 | 40.00 | **60.91** |

a higher budget of $\epsilon + s\_size$, where $\epsilon$ is the allowed perturbation budget ($\approx 0.031$ for CIFAR-10) and $s\_size$ is the nudging used by ECAC as defense (0.02 for CIFAR-10). The idea is to mislead ECAC into nudging the input towards an incorrect class. Once the vulnerable class is identified, we use a targeted PGD attack towards that class using $jointM$. Crucially, after each PGD iteration, we query the deployed ECAC model to check if the attack succeeded. If affirmative, we retain the adversarial example. We present the algorithm in the Appendix as Algorithm 2. Results are presented in Table 5. Our empirical results underscore that ECAC upholds a significant degree of accuracy in a black-box context where the attacker knows the defense strategy, thereby attesting to its robustness.

We also notice that the accuracy of the surrogate ECAC model, which is under a white-box attack, does not fall to zero but remains significantly higher at $40\%$. This is because of the use of *strongM*.

**Results on ViT architecture:** To demonstrate the robustness and applicability of ECAC across diverse model architectures, we conducted experiments using Vision Transformer (ViT) models on CIFAR-10. We trained the base models using the code provided by Mo et al. (2022). The results show that ECAC significantly improves the robustness of $strongM$ in all three settings, demonstrating its effectiveness with different architectures.

Table 6: ECAC CIFAR-10 accuracy with ViT architecture.

| Models (ViT) | Nat' | Square (100/2.5k) | RayS (1k/10k) | SPSA |
|---|---|---|---|---|
| Natural | **91.8** | 42.5/00.1 | 18.7/00.3 | 06.6 |
| SAT | 76.4 | 65.3/52.7 | 59.5/51.2 | 64.8 |
| ECAC-SAT | 80.1 | **75.8/74.4** | **61.8/58.6** | **76.7** |
| TRADES | 80.6 | 69.7/56.4 | 63.7/53.8 | 69.6 |
| ECAC-TRADES | 84.8 | **79.9/77.9** | **65.7/61.5** | **80.3** |

### 4.3 ABLATION STUDY

We performed ablation studies by varying different parameters of the model, including hyperparameters $s\_size$ and $n\_step$, giving different weights to $strongM$ and $weakM$ in constructing $jointM$, and using different adversarially trained models as $strongM$. Due to space constraints, below we only present the study on varying $s\_size$ and present rest of the results in Appendix B.2. We observed that increasing $s\_size$, the amount of corrective nudging done by ECAC, boosts accuracy for SQA attacks but decreases transfer attack accuracy (4.3.1). Increasing $n\_step$ improves natural accuracy but reduces accuracy on SQA attacks (B.2.1). We also find that equally weighing $strongM$ and $weakM$ in constructing $jointM$ works reasonably well against various attacks (B.2.2).

### 4.3.1 Effect of Varying $s\_size$

We finalized the value of $s\_size$ as $0.02$ when $\epsilon = 8/255$. This parameter was chosen heuristically to achieve a good trade-off between adaptive attack accuracy (i.e., potential vulnerability of the model) and black-box attacks. In Table 7, we present the accuracy for Natural, Square, and RayS attacks for different parameter values.

Table 7: Effect on ECAC accuracy for different values of $s\_size$, under various attacks. The results are for the first 1000 samples of CIFAR-10.

| Accuracy | Value for $s\_size$ | | | | |
|---|---|---|---|---|---|
| | 0.015 | 0.018 | 0.020 | 0.022 | 0.025 |
| Natural | 88.60 | 88.70 | 88.70 | 89.10 | **89.40** |
| Square (1k iterations) | 81.30 | 81.60 | 81.40 | **82.80** | 82.00 |
| RayS (1k iterations) | 72.00 | **72.30** | 72.00 | 72.00 | 71.30 |
| Transfer (using $jointM$) | **69.60** | 68.00 | 67.30 | 66.70 | 65.30 |

We observe that increasing the value of $s\_size$ may yield better results for SQA attacks (Square attack in the table). However, this makes the model more vulnerable to transfer attacks.

### 4.4 Limitations

Our defense is limited to black-box setting, which includes decision based as well as transfer attacks. Thus, the method is useful in many realistic scenarios where attackers lack complete access to deployed models. Further the model retains significant robustness even udner white-box attack.

Another weakness of the ECAC model is that it takes more time to make predictions than a simple feed-forward model. A simple ResNet-18 model takes $\approx 1.6$ seconds to classify the entire CIFAR-10 test set, with a batch size of 500 using an RTX-2080 Ti graphics card, whereas ECAC takes $\approx 8.3$ seconds. This is because it needs to make an extra forward and an extra backward pass. However, as observed, the feed-forward part of biological perception also seems vulnerable to adversarial attacks and needs significantly more time to make a correct prediction. We believe that this weakness could be manageable for many safety-critical applications.

## 5 Conclusion and Discussion

In this work, inspired by biological perception, we argue for the need to go beyond simple feed-forward networks for adversarial robustness. We introduce a novel approach called ECAC, which combines naturally and adversarially trained models to enhance robustness. ECAC uses the disagreement between the two models as an oracle to nudge the input toward the correct class, simulating an error-correction mechanism. To the best of our knowledge, this is the first attempt in this direction. We find that this technique significantly boosts black-box robustness, making it appealing for real-world applications where attackers typically have limited access to deployed models.

This work highlights the potential of error-correction mechanisms to enhance adversarial robustness, suggesting promising future research directions. One idea is to use a generative model to create an image and then compare its similarity to the original image using, for instance, $\ell_2$ distance. This approach, however, requires generating high-quality images, which can be challenging.

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

# A  MODEL AGREEMENT AND CORRECT PREDICTION

We work with the assumption that when the two models agree, their agreed prediction tends to be correct. While this is intuitive, as the models are trained to make the correct prediction, we further analyze how and when this assumption holds.

Our assumption is based on the intuition that the chance of *StrongM* and *WeakM* predicting the same wrong class is low. Formally, let the accuracy of *StrongM* be $p_s$ and *WeakM* $p_w$, their predictions be $S_p$ and $W_p$, and $y$ be the correct prediction. Assume their predictions are independent and equally likely to predict any of the wrong labels, then

$$P(S_p = y')_{y' \neq y} = \frac{(1 - p_s)}{(C - 1)}, \tag{2}$$

$$P(W_p = y')_{y' \neq y} = \frac{(1 - p_w)}{(C - 1)} \tag{3}$$

where $C$ is the number of classes. We have:

$$\frac{P(correct|agree)}{P(incorrect|agree)} = \frac{P(correct, agree)}{P(incorrect, agree)} \tag{4}$$

$$= \frac{P(S_p = y, W_p = y)}{\sum_{y' \neq y} P(S_p = y', W_p = y')} \tag{5}$$

$$= (C - 1)\frac{p_s}{(1 - p_s)}\frac{p_w}{(1 - p_w)}. \tag{6}$$

Therefore, the assumption holds when $p_s$, $p_w > 50\%$ for binary classification and could hold even when $p_s$, $p_w$ is small for multi-class classification.

Further, as noted in the main paper, we experimentally verified this. For CIFAR-10 using the ResNet-18 model, we found that when an adversary is both crafted and nudged using both *strongM* and *weakM* in parallel (i.e., *jointM*), then nudging towards the correct class increases the agreement between the two models' predictions from $34.98\%$ to $49.20\%$, while if they are nudged towards a random class, the agreement decreases to $16.99\%$.

# B  ABLATION STUDY AND DESIGN CHOICES FOR ECAC

We first discuss the design choices, after which we present the ablation studies, where we discuss the tradeoff in varying several parameters.

## B.1  DESIGN CHOICES FOR ECAC

In this section, we discuss how we made decisions regarding several nudging aspects for ECAC. Before nudging, we categorize inputs into the following cases:

1. Both $strongM$ and $weakM$ agree:
   (a) The prediction is correct.
   (b) The prediction is incorrect.
2. The two models disagree:
   (a) Both $strongM$ and $weakM$ predictions are incorrect.
   (b) $strongM$ prediction is correct.
   (c) $weakM$ prediction is correct.

In Table 8 we provide the percentage of natural samples from the CIFAR-10 test-set that fall into each of the cases above.

For case 1, when the two models agree, we do not have a signal to check if the prediction is correct or not. However, in this case, for natural samples, there is less than a $2\%$ chance that the two models

Table 8: Percentage of samples from the CIFAR-10 testset falling into different cases.

| Case | Samples % |
|---|---|
| Case 1 (a) | 81.79 |
| Case 1 (b) | 1.86 |
| Case 2 (a) | 1.48 |
| Case 2 (b) | 1.80 |
| Case 2 (c) | 13.07 |

would disagree. For case 2, however, we can use nudging. Since it is easier to fool $weakM$, as also evident from the results in Table 2, we focus on improving the robustness of $strongM$. In case 2 (a), when both models are making an incorrect prediction nudging may not be that useful. However this is unlikely for natural samples. For case 2 (b), nudging can actually lower the prediction. However, in a black-box setting it would be very difficult for an attacker to achieve this case. Also, by default, only $1.8\%$ of samples fall into this category. Finally for case 2 (c), which is the second-highest category with $13\%$ of samples, ECAC fixes the prediction.

## B.2 ABLATION STUDY

### B.2.1 EFFECT OF VARYING $n\_steps$

The two parameters, $n\_steps$ and $s\_size$, determine the amount of corrective nudging done by ECAC. We present the results of varying $n\_steps$ for three settings in Table 9. The results are for the ResNet-18 model, with the Square attack run for 2.5k iterations. In all settings, each step size was chosen so that the combined $s\_size$ is close to 0.02. Before nudging the input, we clipped the nudging to 0.02 for a fair comparison.

Table 9: Effect on ECAC accuracy for different values of $n\_steps$, where total perturbation by nudging was clipped at 0.02. The results are for the first 1000 samples of CIFAR-10.

| $n\_steps$ | Size of each step | Natural Accuracy | Square Attack Accuracy | Square Attack Time (sec) |
|---|---|---|---|---|
| 1 | 0.020 | 87.8 | 82.4 | 509.6 |
| 2 | 0.011 | 89.3 | 79.6 | 814.4 |
| 3 | 0.007 | 89.3 | 78.6 | 1123.1 |

From the ablation study of $n\_steps$, we infer the following: Incremental steps appear to be marginally more effective for nudging towards the goal than a single step. This is not surprising, as iterative attacks like PGD are more effective than single-step attacks like FGSM. With unperturbed input, a higher step count correlates with increased natural accuracy, as it corrects more samples incorrectly classified by strongM. However, with the Square attack, there is a slight decrease in accuracy, possibly because the attack misleads weakM more readily, inadvertently steering strongM toward incorrect classifications.

As anticipated, a greater number of steps result in longer processing times, with the duration increasing linearly. Therefore, we recommend setting $n\_steps$ to 1 for optimal efficiency.

### B.2.2 CHANGING CONTRIBUTION OF TWO MODELS IN $jointM$

To define $jointM$, we added the loss terms for the two models, giving equal weightage to both the models. However, a weighted sum of the two loss terms, where the models contribute differently, could lead to better robustness. To investigate this, we introduced a new weighting parameter, $\alpha$. The loss for $jointM$ is defined as: $\alpha \times weakM$ loss + $(1 - \alpha) \times strongM$ loss. We present the results in Table 10 for several attacks.

Table 10: Effect on ECAC accuracy on CIFAR-10 for different value of $\alpha$ as used to define $jointM$. ECAC uses WideResNet-28-10 models, while surrogate models uses ResNet-18 architecture.

| $alpha$ | Natural Accuracy | Square Attack Accuracy | Transfer Attack Acc' using surrog' | | |
|---|---|---|---|---|---|
| | | | $strongM$ | $weakM$ | $jointM$ |
| 0.0 | 91.8 | 80.3 | 70.84 | 77.88 | 70.76 |
| 0.1 | 91.4 | 82.7 | 69.88 | 78.94 | 71.49 |
| 0.2 | 91.0 | 84.0 | 69.48 | 78.94 | 71.80 |
| 0.3 | 91.1 | 84.3 | 69.51 | 79.31 | 71.75 |
| 0.4 | 90.9 | 84.8 | 69.38 | 79.54 | 71.79 |
| 0.5 | 90.9 | 84.5 | 69.31 | 79.45 | 72.20 |
| 0.6 | 90.4 | 85.1 | 68.78 | 79.49 | 72.12 |
| 0.7 | 90.3 | 85.5 | 68.72 | 79.66 | 72.41 |
| 0.8 | 90.3 | 85.2 | 68.56 | 79.77 | 72.50 |
| 0.9 | 89.8 | 84.4 | 68.78 | 80.11 | 72.44 |
| 1.0 | 86.6 | 81.7 | 65.62 | 84.44 | 77.11 |

We observe a trade-off among several accuracies, which aligns with our expectations. When $\alpha$ is low, nudging is directed more toward $strongM$ features, as the contribution from $weakM$ loss is minimal. Consequently, post nudging, $strongM$ predictions are expected to align more closely with $weakM$ predictions, as we are nudging toward what $weakM$ foresees. Hence, natural accuracy is higher when $\alpha$ is low because $weakM$ predicts natural samples more accurately. Similarly, for transfer attacks where adversaries are crafted using $strongM$ alone, $weakM$'s prediction on them would be high; thus, the accuracy is higher as well. Conversely, when adversaries are crafted using $weakM$ or $jointM$, $weakM$'s accuracy is lower. Therefore, in these instances, ECAC accuracy is low when $\alpha$ is low and increases monotonically.

For the SQUARE attack, since perturbations are random, both $strongM$ and $weakM$ accuracies decreases. Hence, it is necessary to stabilize features for both models. We observe higher accuracy around the mid-value of $\alpha$. Overall, we conclude that an $\alpha$ of 0.5 is a reasonably effective choice.

### B.2.3 CHANGING *strongM*

ECAC relies on the distinct features used by $weakM$ and $strongM$. There are several ways to achieve $strongM$, making the choice of $strongM$ another hyper-parameter. For a fair comparison with the baseline, AAA (Chen et al., 2022), we only used models trained on WideResNet-28-10. In addition to SAT (Madry et al., 2018), we used TRADES (Zhang et al., 2019), AWP, AWP_E (Wu et al., 2020), and WANG23 (Wang et al., 2023) as $strongM$. We obtained all the pre-trained models from the official repository of Croce et al. (2021), except TRADES, which we trained locally using the same parameters as SAT.

In table 11 we show how ECAC boosts the robustness of corresponding $strongM$ for different attacks. As noted in section 4.2 using these stronger $strongM$ models, we easily surpass the robustness achieved by the baseline AAA.

### B.2.4 DIFFERENT WAYS OF NUDGING THE INPUT

In this work, we used a simple 1-step targeted PGD attack to nudge the input toward the expected correct class. The nudging is the same for all samples. A potentially better approach would be to use an adaptive technique for nudging, such as AutoPGD (Croce & Hein, 2020b). AutoPGD is an advanced iteration of the PGD attack that incorporates momentum from previous iterations and adaptively adjusts the step size and the number of iterations based on variations in the objective function's value. Implementing AutoPGD for nudging—replacing our current heuristic approach to determine nudging parameters, as discussed in Section 4.1—could refine our methodology by tailoring the nudging process to individual samples, potentially enhancing accuracy. However, this adaptation would necessitate multiple iterative steps, significantly increasing the computation time required for ECAC classification. To balance efficacy and efficiency, our current implementation

Table 11: ECAC performance compared to baselines on SQA attacks for the CIFAR-10 dataset with a perturbation budget of $\ell_\infty = \frac{8}{255}$ (queries = 100/2500). WideResNet-28-10 is used for all models.

| Defense Methodology | Nat' Acc' | Accuracy on SQA Attack (# queries = 100/2500) | | | | |
|---|---|---|---|---|---|---|
| | | Square | SignHunter | SimBA | NES | Bandit |
| SAT | 85.83 | 76.9/60.5 | 74.9/56.6 | 84.1/80.4 | 83.3/75.4 | 78.7/66.2 |
| ECAC-SAT | 90.30 | 85.7/84.3 | 80.5/79.4 | 86.0/83.6 | 87.3/73.2 | 85.0/81.9 |
| TRADES | 86.40 | 77.1/61.2 | 74.9/57.0 | 86.2/82.6 | 85.4/74.8 | 80.3/66.2 |
| ECAC-TRADES | 91.65 | 87.4/85.8 | 81.0/79.0 | 86.6/85.8 | 87.9/74.7 | 85.7/83.0 |
| AWP | 85.36 | 75.9/62.7 | 74.0/60.0 | 84.1/80.4 | 83.4/75.2 | 79.1/68.6 |
| ECAC-AWP | 90.00 | 86.9/85.0 | 79.7/77.7 | 86.4/84.9 | 87.0/75.1 | 85.1/82.5 |
| AWP_E | 88.25 | 81.3/67.8 | 79.5/63.4 | 87.2/84.4 | 86.9/79.9 | 83.4/72.5 |
| ECAC-AWP_E | 91.80 | 87.8/87.5 | 83.8/82.5 | 88.0/85.8 | 88.4/77.5 | 86.0/83.7 |
| WANG23 | 92.44 | 86.5/75.5 | 85.0/71.6 | 92.1/89.1 | 91.5/84.8 | 87.8/79.7 |
| ECAC-WANG23 | 94.40 | 91.4/90.9 | 87.2/85.6 | 91.0/89.5 | 92.1/81.6 | 89.8/88.0 |

Table 12: ECAC performance compared to baselines on SQA attacks for ImageNet dataset, with a perturbation budget of: $\ell_\infty = \frac{4}{255}$ (#query = 100/2500). WideResNet-50 is used for all models.

| Attack | Undefended | SAT | RND | AAA-Linear | ECAC (Ours) |
|---|---|---|---|---|---|
| ACC(%) | **78.48** | 68.46 | 75.32 | **78.48** | 72.35 |
| Square | 55.40/10.90 | 61.90/54.40 | 58.67/50.54 | 64.35/63.96 | **67.05/64.95** |
| SignHunter | 62.25/17.30 | 62.65/58.25 | 59.36/52.98 | **71.75 /71.25** | 67.25/64.80 |
| SimBA | 70.65/57.35 | 66.40/64.80 | 66.36/63.27 | 70.80/66.20 | **72.75/69.90** |
| NES | 76.15/59.35 | 67.15/64.65 | 71.33/66.05 | **76.60/70.25** | 70.80/66.25 |
| Bandit | 62.60/27.65 | 64.70/59.45 | 65.15/61.38 | **69.70/69.20** | 69.10/67.95 |

employs a simplified, single-step nudging approach that has demonstrated satisfactory performance in practical applications.

# C  MORE RESULTS

We present additional results in this section, which we couldn't include in the main paper due to space constraints.

## C.1  ACCURACY OF IMAGENET FOR SQA ATTACKS

We present the accuracy of ImageNet for SQA attacks in Table 12.

## C.2  TRANSFER ATTACK FOR IMAGENET

In this section, we present the transfer attack for ImageNet. Since training ImageNet models is computationally expensive (especially adversarially robust models), we used the same component models (i.e., $strongM$ and $weakM$) that are used for defense. We tabulate our results in Table 13. It is not surprising that AAA accuracy drops to zero when we craft the attack using the same naturally trained model. More importantly, we observe that, although we used the same component models that were used for ECAC, the accuracy for ECAC remains considerably high.

## C.3  ADAPTIVE ATTACKS

In this section we provide the algorithm for the adaptive-attack (Algorithm 2).

Table 13: Transfer accuracy of adversaries generated for ImageNet. We used the same component models that are used in the defense.

| Surrogate M (WRN-50) | Undefended | SAT | AAA | ECAC |
|---|---|---|---|---|
| | (WideResNet-50) | | | |
| AT | 67.05 | 40.65 | 67.05 | 48.95 |
| Nat' trained | 00.01 | 67.15 | **00.01** | 61.90 |
| $jointM$ | 00.00 | 59.10 | **00.00** | 55.70 |

---

**Algorithm 2** Adaptive attack

---

**Inputs:**

- $(x, y)$: Input and label pair.
- $surrStrongM$, $surrogWeakM$: Surrogate models.
- $ECAC$: Deployed ECAC model.
- $s\_size$, $\epsilon$: parameters used in ECAC-defense and perturbation budget.
- $pgd\_itrs$, $pgd\_s\_size$: Iterations used for PGD attack.
- $pgdAtk(input, label, model, pert\_bdgt, pgd\_itrs, pgd\_s\_size, do\_t)$:  # A function to do pgd attack. $do\_t$ is a binary variable which indicates if attack is targeted (if $True$) or not.

**Output:** $x'$ such that $ECAC(x') \neq y$ or $FAILURE$ if no such $x'$ is found.

1: $x_t \leftarrow pgdAtk(x, y, surrStrongM, \epsilon + s\_size, pgd\_itrs, pgd\_s\_size, False)$.
2: $y_t \leftarrow argmax(surrStrongM(x_t))$
3: $do\_t \leftarrow True$
4: **if** $y_t == y$ **then**
5:    $do\_t \leftarrow False$.
6: **end if**
7: $x' \leftarrow x$
8: $jointM \leftarrow$ Combine $surrStrongM$ and $surrogWeakM$.
9: **for** $itr = 1$ to $pgd\_itrs$ **do**
10:    $x' \leftarrow pgdAtk(x', y_t, jointM, \epsilon, 1, pgd\_s\_size, do\_t)$.
11:    **if** $argmax(ECAC(x')) \neq y$ **then**
12:       $Return\ x'$
13:    **end if**
14: **end for**
15: $Return\ FAILURE$

---

Our initial phase identifies the most susceptible class for $surrogStrongM$ within perturbation budget of $\epsilon + s\_size$ (Steps 1 and 2). Subsequent to this identification, a targeted PGD attack is executed towards this identified class (with budget of only $\epsilon$). Conversely, should this approach not yield success, we revert to an untargeted PGD attack (Steps 3 to 6), although this scenario seldom deceives ECAC, as $strongM$ is not fooled even at a higher perturbation. During the terminal iterative process (Steps 9 to 14), the attack is driven by surrogate-$jointM$, where each update to the input is evaluated against the operational ECAC model. The process culminates successfully with the delivery of the altered sample upon a successful deception or, failing all iterations, culminates in a declaration of failure (Step 15).

We also point out that it is possible to merge AAA-Linear and RND defenses with ECAC. For the RND method, we can add noise to the input before classification, and we can apply the AAA-Linear technique to the final output. We leave this as a future direction.

## D  ADVERSARIAL ATTACKS AND DEFENSES

We discuss the existing literature on Adversarial attacks and defense in this section.

**Adversarial attacks:** Popular white-box attacks include FGSM (Goodfellow et al., 2014), PGD (and targeted-PGD) (Kurakin et al., 2018; Madry et al., 2018), and its variants like AutoPGD (APGD) (Croce & Hein, 2020b), FAB (Croce & Hein, 2020a), and C&W (Carlini & Wagner, 2017). These are the strongest attacks as the adversary has complete access to the model, including its parameters.

In the real-world settings, attackers only have limited access to the model. If adversaries have access to the model's confidence for each of the predicted class, they can progressively search for a better adversary, called Score-based Query Attacks (SQA). Square attack (Andriushchenko et al., 2020) is arguably the most popular attack of this kind. Other popular SQA attacks are Bandit (Ilyas et al., 2018b), SimBA (Simple Black Box attack) (Guo et al., 2019), ZOO (Chen et al., 2017), SignHunter (Al-Dujaili & O'Reilly, 2019), NES (Ilyas et al., 2018a) and others (Cheng et al., 2019), (Papernot et al., 2017).

In hard-label black-box attacks, also known as decision attacks, an attacker only knows the final class predicted by the model. SPSA (Uesato et al., 2018), HopSkipJump (Chen et al., 2020), RayS (Chen & Gu, 2020), and a few other (Ma et al., 2021; Shukla et al., 2021; Cheng et al., 2018; Brendel et al., 2018) attacks have been developed for this setting.

In transfer attack setting, an attacker has access to a surrogate model and craft adversarial attacks using white-box attacks on the surrogate model. Due to transferability of the adversarial examples (Szegedy et al., 2014), such attacks often transfer to the adversary. Transfer attacks, when coupled with above mentioned decision attacks can significantly reduce the number of queries needed by the decision attacks. Sitawarin et al. (2024) further underscores the relevance of defenses against transfer attacks. These attacks are most successful if surrogate model is trained on the same data and has same architecture.

**Adversarial defense:** Defenses based on Adversarial Training (AT), which incorporates adversarial samples during training, have been most successful so far. SAT (Madry et al., 2018) and TRADES (Zhang et al., 2019) are the most popular defenses and several other defenses extend these methods in some way: e.g. MART (Wang et al., 2019) GAIRAT (Zhang et al., 2020), HE (Pang et al., 2020; Fakorede et al., 2023), MAIL (Liu et al., 2021), AWP (Wu et al., 2020; Yu et al., 2022). For an exhaustive survey of the field kindly refer to (Akhtar et al., 2021).

# E  MODEL SOURCE AND TRAINING DETAILS USED FOR DEFENSE

Table 14: Source for different WideResNet models. * indicates that the models are obtained from RobustBench (Croce et al., 2021) and the corresponding source column contains the Model-ID

| Dataset | $strongM/weakM$ | Model Architecture | Model-Source |
|---|---|---|---|
| CIFAR-10 | $weakM$ | WideResNet-28-10 | Standard* |
| | $strongM$-SAT | | Trained locally |
| | $strongM$-TRADES | | Trained locally |
| | $strongM$-AWP | | Wu2020Adversarial* |
| | $strongM$-AWP_E | | Wu2020Adversarial_extra* |
| | $strongM$-WANG23 | | Wang2023Better_WRN-28-10* |
| ImageNet | $weakM$ | WideResNet-50 | From PyTorch: wide_resnet50_2 |
| | $strongM$-SAT | | Salman2020Do_50_2* |

We provide the source of the WideResNet models in Table 14. For ResNet-18 and for cases where the corresponding model is not present on RobustBench (i.e., $strongM$ for CIFAR-10), we trained the model locally. We used Madry's et al. (2018) method to train all the adversarially robust models for CIFAR-10, which are used as $strongM$. In line with the settings used in the literature (Wang et al., 2019; Liu et al., 2021), all the base models (i.e., those included in Table 3 as well) have been trained for 120 epochs using mini-batch gradient descent with an initial learning rate of 0.01 (0.1 for WideResNet), which was decayed by a factor of 10 at epoch 75, 90 and 100. The values for other

hyper-parameters are weight decay: 0.0035 (0.0007 for WideResNet), momentum: 0.9, and batch size: 128.

Table 15: Hyper-parameters as used for SQA attacks

| Method | Hyperparameter | CIFAR-10 | ImageNet |
|--------|----------------|----------|----------|
| Square | p (fraction of pixels changed every iteration) | 0.05 | 0.3 |
| SignHunter | $\delta$ (finite difference probe) | 8([0, 255]) | 0.1([0, 1]) |
| SimBA | $d$ (dimensionality of 2D frequency space) | 32 | 32 |
| | $order$ (order of coordinate selection) | random | random |
| | $\epsilon$ (step size per iteration) | $\frac{8}{255}$ | $\frac{4}{255}$ |
| NES | $\delta$ (finite difference probe) | 2.55 | 0.1 |
| | $\eta$ (image $l_p$ learning rate) | 2 | 0.002 |
| | $q$ (# finite difference estimations / step) | 20 | 100 |
| Bandit | $\delta$ (finite difference probe) | 2.55 | 0.1 |
| | $\eta$ (image $l_p$ learning rate) | 2.55 | 0.01 |
| | $\tau$ (online convex optimization learning rate) | 0.1 | 0.01 |
| | $Tile\ size$ (data-dependent prior) | 20 | 50 |
| | $\zeta$ (bandit exploration) | 0.1 | 0.1 |

## F    PARAMETERS USED FOR SQA ATTACKS

We used the same parameters as used by AAA defense for most of the attacks. We adapted the code from BlackBoxBench (`https://github.com/adverML/BlackboxBench`), except for SimBA, which we discuss in the next paragraph. Further we provide the JSON files that have the values of parameters we used for the attacks. The details of the parameters have been compiled in Table 15. For the square attack, for which we used code provided by auto attack, we used p_init = 0.05 for CIFAR-10 and 0.3 for ImageNet.

For the SimBA attack (Guo et al., 2019), we observed that if we change one pixel at a time, both AAA-Linear and ECAC do not get fooled. Although it does bring the accuracy for the naturally trained model very low. Thus, we used the SimBA-DCT attack. This is not as effective against naturally trained models but is more effective for adaptive attacks like AAA-Linear and ECAC. We adapted the code from the simple-blackbox-attack repository (`https://github.com/cg563/simple-blackbox-attack`). We also modified the DCT attack such that the minimum perturbation is scaled to the allowed maximum perturbation.

