# OpenReview forum: "Error Correcting by Agreement Checking for Adversarial Robustness against Black-box Attacks"
_ICLR.cc/2025/Conference — ICLR 2025 Conference Withdrawn Submission_

### Official Review · Reviewer_jyyf · 2024-11-02

**Soundness:** 3
**Presentation:** 3
**Contribution:** 3
**Rating:** 6
**Confidence:** 5

**Summary:**

In this paper, the authors propose a novel defense strategy named Error Correcting by Agreement Checking (ECAC) against black-box adversarial attacks. The authors recognize the distinct feature sets used by naturally and adversarially trained models for classification. ECAC exploits this disparity by nudging the input towards the prediction of the naturally trained model unless it leads to a prediction disagreement between the two models. The effectiveness of ECAC is demonstrated through comprehensive experiments on various datasets and architectures.

**Strengths:**

1.	The adversarial defense method proposed in this paper, which capitalizes on the disparities in the outputs of adversarially trained models and naturally trained models, offers a novel perspective for this field.
2.	This paper provides sufficient insights regarding the proposed ECAC method and clearly elaborates on the motivation behind it.
3.	This paper is easy to follow.

**Weaknesses:**

1.	Compared with traditional adversarial training methods, the ECAC method requires the reliance on an additional normally trained model, which incurs additional resource consumption and computational overhead. The authors need provide the results regarding the additional time expenditure.
2.	The authors claim in the title that the proposed method is targeted at black-box adversarial attacks. However, it seems to me that each module of the method does not have a customized design for black-box adversarial attacks. It is recommended that the authors supplement relevant explanations in the method section or directly consider revising the title.
3.	The experiments in this paper do not provide error bars or results from experiments with different random seeds, which raises my concerns about the validity and stability of the experimental outcomes.

**Questions:**

See Weakness

---

### Official Review · Reviewer_ebhx · 2024-11-02

**Soundness:** 3
**Presentation:** 2
**Contribution:** 2
**Rating:** 3
**Confidence:** 4

**Summary:**

This paper explores a new defense method named Error Correcting by Agreement Checking (ECAC) against black-box adversarial attacks. ECAC leverages the difference between natural and adversarially trained models to move the input toward the prediction of the naturally trained model before outputting predictions. This leads to an error correction mechanism that maintains significant robustness even in white-box attacks.

**Strengths:**

- Originality: the paper considers an effective approach to defend against black-box attacks by correcting errors based on agreement checking, which sounds intuitively reasonable.
- Clarity: The motivation of the paper is clear and convincing.
- Significant: The experiments are done against popular black-box attacks, and the empirical improvements compared to baselines seem significant. A minimal performance tradeoff is desirable.

**Weaknesses:**

My main concerns are:
1. Many statements and arguments have few supporting data and references.
2. The paper does not flow because some parts of the paper are difficult to follow and confusing.
3. The experiment setup is not well written, and the results of the experiments are not well presented.
4. The proposed are developed based on an assumption without theory. It could be a flaw.

Please see the section on questions for a list of my specific concerns.

**Questions:**

## Main questions:
1. The observation of the authors about the shape-like and texture-like features that are orthogonal is not obvious. The results from Table 1 do not demonstrate this explicitly and intuitively. I suggest the authors demonstrate a more intuitive or rigorous analysis to support this observation and statement.
2. “$l_\infty$ norm attacks are much stronger” is vague and incomplete. Does it mean $l_\infty$ norm attacks are much stronger than other $l_p$ norm attacks? If so, it might be a flawed statement if the authors do not provide supporting data or references. I argue that different $l_0$ and $l_2$ norm attacks are stealthy and strong and encourage the authors to evaluate the proposed method and other defense mechanisms against both $l_0$ and $l_2$norm attacks, such as SquareAttack and SignHunter ($l_2$) and SparseRS ($l_0$) [1] or BruSLiAttack ($l_0$) [2].
3. Can the authors provide a thorough analysis on "not increasing the correct decision area around an input" results in "exposed to transfer attacks"? In general, increasing the correct decision area around an input enhances the robustness against all adversarial attacks, including transfer attacks. Specifically, to transfer attacks, some research [3, 4] pointed out that transfer-based attacks’ success relies on training conditions and constraints in crafting adversarial examples as well as the similarity between the surrogates and target models.
4. For Table 1, what is transfer accuracy? Why is the transfer accuracy from the same model not high e.g. the transfer accuracy from SAT to SAT or TRADES to TRADES is lower than 55%?
5. Texture-like features are easier to remove by adversarial training. I encourage the authors to provide a reference to support this statement. Similarly, the authors should provide a reference for the statement about shape-aware features that can be learned by adversarially trained models.
6. The assumption that “the agreed predictions between weakM and strongM tend to be correct” is weak and does not reflect the truth. The assumption is only true if both weakM and strongM are trained well and achieve high clean accuracy because the correct predictions of both of them are expected high. Additionally, this assumption depends mainly on how large the overlapped regions between the decision boundaries of weakM and strongM are and how large the correct part of these regions is. This is because the predictions of all samples in the overlapped regions made by weakM and strongM are the same and they can be correct or incorrect. WeakM and strongM are different so their decision boundaries are different. However, there is no way to determine how large the correct part of the overlapped regions is and there is no guarantee that the assumption hold during the attack. The analysis in Appendix A only holds for clean images e.g. images in an evaluation set, but not the entire data distribution, as we do not have that. Since the proposed method is designed based on a weak assumption, it is not robust. However, if the authors believe your assumption is strong, I encourage the authors to provide a theoretical analysis.
7. The explanation and demonstration of the error correction mechanism in Figure 3 and Section 3.3.1 are confusing and do not make sense. Even though they aim to illustrate how ECAC still obtain a high accuracy while the accuracy of strongM drops, it does not fundamentally show how the correct mechanism works against black-box attacks. Instead of the accuracy, I suggest illustrating the difference in the confidence scores between strongM and ECAC e.g. strongM has reduced scores while ECAC maintains high confidence scores due to the effect of error correction.
8. The authors should provide evidence or a reference to support the statement “a random perturbation will rarely cause both models to fail”. This relies on how similar the decision boundaries of both models are and mainly holds if both models are very diverse.
9. I was confused about how SPSA was a hard-label black-box attack. Since the authors mentioned gradient estimation attacks in hard-label settings, it would be more appropriate if the authors discussed HopSkipJump or SignOPT [5]. I suggest the authors should evaluate your methods and others under one of these hard-label attacks.
10. The effect of ECAC relies on the amount of nudging and perturbation budget (lines 317, 318). But in practice, the perturbation budget is unavailable to defenders. The results in table 7 show that ECAC can be tuned for a given perturbation budget so ECAC is ineffective across different perturbation budgets.
11. I encourage the authors to evaluate ECAC against L2 attacks as in [6]. Furthermore, To demonstrate the effectiveness of ECAC against a wide range of black-box attacks, the authors should illustrate the performance of ECAC against L0 attacks [1, 2].
12. Why did the authors use different models, Resnet-18 and WideResNet-28-10, in different experiments? Why did the authors conduct different experiments with ViT against decision-based attacks and did not explain clearly what strongM used in the experiments?
13. I was confused about the term "nudging towards the correct class". As shown in Equation 1, ECAC nudges along a direction between the gradient direction of the loss for weakM and strongM. This direction is not always towards the correct class. The authors should provide a better explanation for this point.

## Minor
1. Section 2.2, the title is adversarial attacks, but it does not say anything about adversarial attacks. I suggest the author should bring some related work in the Appendix to this section. To make it more complete, the authors should discuss $l_0$ norm attack (sparse attacks both score-based and decision-based attacks).
2. I was confused about Equation 1, particularly the notation of $loss(x’^t_i, y^o_i)$ and product $\prod$. How does $loss(x’^t_i, y^o_i)$ represent the summation of the CE loss incurred by weakM and strongM? It would help to explain or provide a complete formulation of this $loss(.)$. What is the role of the product $\prod$ in Equation 1?
3. Figure 3, the authors should add titles to x/y axis.
4. The authors should provide a reference for numbers mentioned in the paper e.g. “nearly 10% less effective” (line 302), “drop from 83.5% to 71.8%” (line 309).
5. The setup section is not clearly presented and lacks important information. The authors should clearly explain whether defense methods are evaluated on the entire evaluation set or a subset from CIFAR-10 and ImageNet. All attack and defense methods should be clearly mentioned. What metric is used to measure the performance of defenses? What is the query budget used by attacks? Describe how to train and obtain strongM model or make a reference to where you discuss it.
6. What does queries = 100/2500 mean? I was confused about the results in Table 2 e.g. 83.4/79.8 or 91.4/90.9. How should I understand these results?

[1] Maksym Andriushchenko, Francesco Croce, Nicolas Flammarion, and Matthias Hein. Square Attack: a query-efficient black-box adversarial attack via random search. ECCV, 2020

[2] Quoc Viet Vo, Ehsan Abbasnejad, and Damith C. Ranasinghe. Brusleattack: Query-efficient score-based sparse adversarial attack. ICLR, 2024.

[3] Suya, Fnu, et al. "Sok: Pitfalls in evaluating black-box attacks." 2024 IEEE Conference on Secure and Trustworthy Machine Learning (SaTML). IEEE, 2024.

[4] Pin-Yu Chen, Huan Zhang, Yash Sharma, Jinfeng Yi, and Cho-Jui Hsieh. Zoo: Zeroth order optimization based black-box attacks to deep neural networks without training substitute models. In ACM Workshop on Artificial Intelligence and Security (AISec), pp. 15–26, 2017

[5] M. Cheng, S Singh, P Chen, P.-Y. Chen, H. Yi, J. Zhang, and C.-J. Hsieh. Sign-opt: A query-efficient hard-label adversarial attack. ICLR, 2020

[6] Zeyu Qin, Yanbo Fan, Hongyuan Zha, and Baoyuan Wu. Random noise defense against query-based black-box attacks. NIPS, 2021.

---

> ### Author Response · Authors · 2024-11-14
>
> We are immensely grateful for your time and thorough analysis of our work. We acknowledge several oversights in our writing and sincerely thank you for identifying them. We will address all of these in our next submission.

---

### Official Review · Reviewer_zaEU · 2024-11-04

**Soundness:** 2
**Presentation:** 3
**Contribution:** 2
**Rating:** 5
**Confidence:** 5

**Summary:**

The paper proposes ECAC, a method to improve the robustness of image classifiers against black-box adversarial attacks. ECAC uses a naturally trained and an adversarially trained model: if the predictions of the natural model on the original input and of the robust model on a perturbed version of the input agree, then the latter is returned, otherwise the prediction of the robust model on the original input. This aim at exploiting the fact the natural and robust models rely on different sets of features, and typically attacks do no transfer among these models. In the experiments the proposed method is evaluated against several (query- and decision-based, transfer) black-box attacks, and it is shown to outperform the baselines.

**Strengths:**

- The proposed method is well presented.

- The experimental evaluation is extensive, with many different attacks, including adaptive ones, tested, and shows that ECAC outperforms the baselines. Moreover, the paper includes several ablation studies.

**Weaknesses:**

- The scope of the paper, i.e. robustness to black-box attacks, is quite narrow when compared to existing methods, e.g. the adversarially trained models provide robustness to white-box attacks too.

- ECAC might even have lower robust accuracy than the standalone robust model, as the attacker could exploit the "nudging" mechanism to create perturbations larger that those used for adversarial training (as also shown in Table 7, if I interpret it correctly). The proposed method shares some similarity to techniques like [A], which has been shown ineffective against adaptive attacks [B].

- Some aspects of the evaluation are not convincing.
  - The budget for the black-box attacks might be insufficient: for example, 2500 queries for score-based methods or 10000 for RayS might not be sufficient for the algorithms to converge, as attacks on ECAC are expected to be more challenging to optimize than for standard models.
  - Similarly, transfer attacks use only PGD-20, and from smaller architectures.
  - Every more sophisticated fully black-box attack (Table 2, Table 3, Table 4) yields lower robust accuracy for ECAC than the previous one (and it even lower is for the grey-box setup in Table 5): this gives the general impression that the evaluation might still be improved. Moreover, it's not clear what the white-box robustness of the adversarially trained model is, which would serve as a minimal value upon which ECAC should improve.

- The proposed method notably increases the inference cost.

[A] https://arxiv.org/abs/2106.04938 \
[B] https://arxiv.org/abs/2202.13711

**Questions:**

Overall, it is not clear from the experiments what's the actual gain of the proposed defense compared to e.g. a single adversarially trained model, as there seems to be room for improving the evaluation. Moreover, the approach is limited to black-box attacks, and increases the inference time. It'd be important to address these weaknesses (see detail above) of the proposed method.

---

### Official Review · Reviewer_53d9 · 2024-11-09

**Soundness:** 4
**Presentation:** 4
**Contribution:** 3
**Rating:** 6
**Confidence:** 5

**Summary:**

The paper introduces Error Correcting by Agreement Checking (ECAC), a novel defense method aimed at countering black-box adversarial attacks on machine learning models, inspired by the vulnerability of the initial feed-forward phase in biological perception. Unlike conventional methods, ECAC leverages the distinct features used by naturally and adversarially trained models, shifting inputs towards the naturally trained model’s predictions while checking for prediction alignment between the two models. This approach effectively defends against leading black-box attack types, including score-based, decision-based, and transfer-based attacks, and maintains robustness even under full adversarial model access. Extensive experiments on CIFAR and ImageNet datasets using ResNet and ViT architectures confirm ECAC’s efficacy in improving model robustness.

**Strengths:**

1. The paper presents a novel method for defending image classifiers against evasion attacks using an error-correction mechanism.
2. The empirical results are comprehensive, using diverse attacks, datasets, and models. The results demonstrate notable improvement in robustness against black-box attacks over the baselines.
3. Adaptive attacks are discussed and corresponding results are presented in the paper.
4. The design choices are adequately discussed and backed up with ablation studies.

**Weaknesses:**

1. While authors have expended adequate efforts towards designing an adaptive attacks against their defense, the execution of the attacks is lacking. The authors perform this attack in a black-box setting using a surrogate model (i.e., a transfer-based black-box attack). This is one of the weaker class of black-box attacks. While staying within the black-box threat model, the attack can potentially be made stronger by using the Square attack alogrithm, but replacing the cross-entropy loss with the adaptive attack loss (targeted loss on jointM). Additionally, it is critical to report the results of an adaptive attack in a complete white-box setting to establish the lower-bound of robustness of the defense. However, such results are not reported.
2. Authors use a much smaller value for p_init in Square attack than what is recommended, making the attack much weaker. Authors justify this choice by stating that the prior work they compare against (AAA-line) performs much worse for p_init = 0.8 (recommended value) and therefore they instead use p_init = 0.05, the value used in the AAA-linear paper. Considering that empirical defenses quantify robustness using attacks, it is imperative that the strongest possible attack be used to get an accurate measurement of robustness. As such, I think the more valuable comparison is the one using p_init=0.8. And these results should be added to the main paper to improve empirical soundness.

**Questions:**

1. What is the success rate of a white-box targeted attack on jointM? How do these adversarial examples fare against the E2E error-correction pipeline?
2. Is the target label in the adaptive attack obtained by attacking the original jointM or the surrogate one?
3. How do the defense methods compare when using Square Attack with p_init = 0.8?
4. L#141: "it remains exposed to transfer attacks.", Are there any results/references to back this up?

---

### Note · Authors · 2024-11-14

**Comment:**

We sincerely thank the reviewer for their invaluable feedback and time. We will incorporate the provided comments for our next submission.

**Withdrawal Confirmation:**

I have read and agree with the venue's withdrawal policy on behalf of myself and my co-authors.